# Fast Inline Microscopic Computational Imaging

**DOI:** 10.3390/s22187038

**Published:** 2022-09-17

**Authors:** Laurin Ginner, Simon Breuss, Lukas Traxler

**Affiliations:** AIT Austrian Institute of Technology GmbH, 1210 Vienna, Austria

**Keywords:** 3D imaging, inline inspection, microscopy

## Abstract

Inline inspection is becoming an essential tool for industrial high-quality production. Unfortunately, the desired acquisition speeds and needs for high-precision imaging are often at the limit of what is physically possible, such as a large field of view at a high spatial resolution. In this paper, a novel light-field and photometry system is presented that addresses this trade off by combining microscopic imaging with special projection optics to generate a parallax effect. This inline microscopic system, together with an image processing pipeline, delivers high-resolution 3D images at high speeds, by using a lateral transport stage changing the optical perspective. Scanning speeds of up to 12 mm/s can be achieved at a depth resolution of 2.8 μm and a lateral sampling of 700 nm/pixel, suitable for inspection in high-quality manufacturing industry.

## 1. Introduction

High quality standards in modern large-volume production call for on-site microscopic inspection. In many applications, such as roll-to-roll manufacturing for printed electronics, photovoltaics, batteries and displays, high resolution 3D imaging is required on site. We previously focused on the fast inline inspection systems with samplings in the range of 50 μm per pixel [1]. In this range, most industrial optical inspection systems rely on structured active illumination, such as laser line triangulation, a pattern projection or different kinds of multi-view stereo systems. For industrial high-quality production, much higher spatial resolutions down to 1 μm are required [2]. In this regime, light diffraction becomes dominant and compromises the overall performance of these techniques, ultimately limiting the spatial resolution that can be achieved. This ultimately limits the applicable resolution ranges of many of these methods. A large variety of microscopic imaging methods exist, but most of them are not well suited for fast inline inspection at high production speeds, due to their time consuming nature (depth stacking, etc.). The aim of this paper is to demonstrate a novel light-field and photometry-based 3D imaging system that is capable of fast acquisition speeds, and hence, inline applications.

### 1.1. State of the Art in 3D, Microscopic Inline Imaging

Laser line triangulation presents one of the most widely used optical inline 3D profilometry-sensing modalities. In the microscopic range, active illumination systems are limited by the smallest possible line focus, also known as the depth of field (DOF) across a certain focal depth $f_{D}=\frac{2\pi w_{0}^{2}}{\lambda_{0}}$, with $w_{0}$ being the beam waist (i.e., lateral resolution) and $\lambda_{0}$ the central wavelength, assuming a Gaussian beam profile. Considering a wavelength of $\lambda_{0}=500$ nm, a line width of 1 μm ($w_{0}=500$ nm) is only possible for a depth of approximately 3–4 μm. Additionally, laser speckling is harder to mitigate at these spatial resolutions. Therefore, these sensors are typically used for samplings down to about 2.5 μm per pixel [3].

Due to this trade-off between spatial resolution and focal depth, multi-view stereo is uncommon for microscopic 3D imaging. The narrow DOF in microscopic imaging can be utilized for focus stacking or DfF (depth from focus), which is a common microscopic imaging modality [4]. Here, the focal distance of the imaging system is changed in small increments, resulting in a stack of images from the same scene with varying focus positions (focus stack). In every image, only a certain depth is in focus. DfF algorithms can retrieve the 3D surface information from such a focus stack. For fast surface inspection, this method presents a large drawback: In every image, only a small portion of the captured image, namely, the intersection of the focal plane with the object’s surface, carries significant information that can be further exploited. A large portion of each image is off focus and does not contain data suitable for further processing. This leads to inefficient usage of the data acquisition bandwidth, in terms of both acquisition time and amount of data to be processed. To overcome this bandwidth limit, fast focal stacking by using event-based cameras has been proposed [4]. For this approach, only those parts of the image scene where there is a change in focus are captured and recorded. Still, DfF is not well suited for industrial inline applications with continuously moving objects, as several images must be acquired with the object in the same position. For 3D imaging with lateral samplings of 1 μm per pixel and smaller, white light interferometry (WLI) and confocal microscopy (CM) are common methods. These methods can deliver incredible depth resolutions (down to sub-μm), but their working principles come at the cost that they are scanning-based, and hence limited in the acquisition times that can be achieved. For WLI, vertical scanning is necessary to probe at various reference path lengths [5], whereas for CM, point-by-point scanning is the usual approach. Both methods are thus difficult to apply in inline applications with continuous object movement.

Fast (up to 100 kHz point scan rate) CM sensors, based on spectral encoding, are available and used for high-speed inline scanning applications [6]. An elegant adaption of chromatic CM, in contrast to the usual point-by-point scanners, is the line scanning sensor, where a full line can be acquired at once [7,8]. This way, CM can be applied to continuously moving objects. However, at present, this technology is only available for lateral samplings down to 2.1 μm point-to-point spacing at a scan rate of 300 Hz at full scan depth and up to 800 Hz for a reduced measurement range [8,9]. The lateral sampling is similar to available laser line triangulation systems but with the benefit of a much better depth resolution.

### 1.2. Aim of This Paper

Previously, we demonstrated that with slight modifications on an industrial inspection microscope, an inline observation of an object from many different observation angles is possible [10]. Similarly to conventional structure-from-motion, which makes use of the parallax changes of 3D objects when observed from different observation angles, 3D information can be retrieved. In this paper, we demonstrate a novel 3D imaging system with a lateral sampling of 700 nm per pixel based on inline light-field and photometry acquisition. Though having introduced the optical principle already in 2019, no theoretical or experimental proof was given. Hence, this paper describes in detail how the depth resolution depending on the aperture size and lateral resolution can be calculated. Additionally, photometric stereo information is used, coping with difficult surface properties and occlusions, to visualize even structures beyond the resolution limit. The system is fully inline capable, and the proposed acquisition principle allows for an acquisition speed of up to 12 mm/s, which would correspond to a line rate of 12 mm/s / 700 nm = 17 kHz or 17,000 3D line profiles per second. The acquisition speed is mainly dependent on the maximum frame rate of the camera (maximum 200 frames per second) and the light being available to illuminate the sample properly. Using the specially designed light sources, depending on the surface properties, exposure time can be cut down to 100 μs. By assuming a constant object transport speed, the computationally expensive pose estimation from structure-from-motion algorithms can be omitted [11]. This way, the depth estimation can be performed as fast as the actual image acquisition, allowing for a novel and fully inline-capable high-speed inspection system. To the best of our knowledge, currently no other 3D imaging technique exists that offers the same high-resolution lateral sampling and acquisition and processing speed as the proposed inline inspection system. In this paper, we will explain in detail the optical configuration and the illumination system. The performance regarding depth resolution of the proposed method is evaluated theoretically and demonstrated experimentally in line with the current recommendations proposed by the "Initiative Fair Data Sheet" [12], which is supported by numerous players from research and industry developing optical surface measurement devices. Additionally, the paper demonstrates a practical industrial use case for inspection of a print plate and security features on banknotes.

## 2. Microscopic Inline Light-Field Acquisition System

In the last few years, an ecosystem of acquisition hardware and software for high-performance inline computational imaging (ICI) was developed by us [1,11]. The basic working principle of this inline 3D imaging system relies on a moving object to be inspected, e.g., an object moving on a production line, which is sequentially captured by a stationary camera. As the object moves, the camera captures the object from a slightly different viewing angle. The observed data are equal to what one would observe with an one-dimensional linear light-field camera array. Via multi-view stereo matching, 3D information can be obtained from this image sequence. For illumination of the object, a special dome light construction was used, utilizing LED spots to illuminate the scene from six different directions with a high luminous flux [13].

### 2.1. Optical Configuration for Inline Light-Field Acquisition

Typical inspection microscopes have telecentric projection properties. This means that the geometric projection center (the entrance pupil) is at an infinite distance. This is useful as the magnification in such a configuration is independent from the object distance. With such a configuration, an object is always seen from the same perspective, without producing any parallax effect, and hence, no stereo matching (see above) can be applied. Hence a hypercentric projection optics is used in to enable stereo matching, see Figure 1.

For the proposed microscopic 3D imaging system, several imaging parameters must be chosen based on the application: sensor size (pixel count in transport direction $n_{t}$ and pixel count normal to the transport direction $n_{n}$), pixel size $p^{\prime }$, magnification *M*, numerical aperture of the objective lens NA, aperture position $d_{a}$, and the aperture diameter *D* that defines the effective $N\;A_{e\;f\;f}$. For a better understanding, we provide practical values for the applications showcased in Section 4.

The sensor size and magnification define the FOV; the pixel size defines the lateral sampling, these design concerns are handled exactly as they would be in a conventional digital microscope. The location of the aperture defines the 3D imaging properties.

As a starting point, a camera sensor and objective lens are chosen to define a suitable FOV and lateral sampling. For fast inline acquisition, a 4 Mpixel CMOS high-speed camera with a frame rate of up to 400 FPS was selected (Bonito CL-400C), with a pixel count of $n_{t}=1726$ and $n_{n}=2320$. The camera’s pixel size is $p^{\prime }=7$ μm. As objective lens, a M=10× long working distance objective with a NA=0.28 in combination with a 200 mm tube lens, was chosen. With that, the lateral sampling *p* is given by $p=p^{\prime }/M=700$ nm. The scan profile width was $p\cdot n_{n}=1.624$ mm.

For choosing appropriate 3D imaging settings, interleaved phenomena have to be understood for a given application:A large aperture diameter *D* increases the lateral optical resolution (see Equation (Equation 1)). A small aperture increases the depth of focus, and thus the depth range (see Equation (Equation 2)).A large aperture position $d_{a}$ decreases the distance between an object and the virtual location of the entrance pupil and thus increases the parallax effect $\alpha_{t}$ and the theoretical z-resolution limit. (see Equation (Equation 3)).A smaller *D* and/or position $d_{a}$ has to be chosen if vignetting occurs due to the finite objective lens diameter (see Figure 2) and defined by the mathematical constraint (Equation (Equation 6)).

Following these considerations, appropriate 3D imaging settings can be derived for the proposed microscopic inspection system: An aperture diameter of D=3.4 mm is used, resulting in an effective numerical aperture of $N\;A_{e\;f\;f}=n\cdot sin\left(atan\left(\frac{D}{2f}\right)\right)=0.085$. Given the diffraction limit (Equation (Equation 1)):(1)$$res_{xy}=\frac{1.22\lambda }{2N\;A_{e\;f\;f}}$$
the lateral optical resolution $res_{xy}$ is 3.9 μm at a 550 nm central wavelength. We confirmed this by imaging an USAF target, being able to visualize segment 7.1. Given the lateral sampling of our image of 700 nm/pixel, an oversampling of approximately 5.6 is achieved.

Considering the feature size for stereo matching (see Section 2.2), the usable depth of field can be estimated by
(2)$$DOF=2\frac{\lambda }{{N\;A_{e\;f\;f}}^{2^{\prime }}}$$
resulting in a depth scan range of approximately 150 μm, which is suitable for our demonstrated use cases.

For the depth resolution $res_{z}$, we assume that a step height can be resolved if the motion parallax between the first and the last view along the transport direction (change in the observation angle of $\pm \alpha_{t}$) is equal to the shift of one pixel *p* (lateral sampling), as depicted in Figure 3. This results in a depth resolution of
(3)$$res_{z}=\frac{p/2}{tan(\alpha_{t})}$$

The maximal projection angle along the transport direction can be calculated using geometric optics (Figure 4): The viewing angle is zero in the center of the FOV and increases off-center. Along the transport direction, the FOV extends to $b=n_{t}/2\cdot p=0.6$ mm away from the optical axis. With this as a center, the observation light cone in the lens plane is displaced by $d_{p}=b\cdot \frac{d_{a}}{f}$. This results in a maximal observation angle along the transport direction:(4)$$\alpha_{t}=atan\left(\frac{d_{p}-b}{f}\right)$$

Combining Equations (Equation 3) and (Equation 4), the depth resolution is given by
(5)$$res_{z}=\frac{p}{2\frac{dp-b}{f}}$$
(6)$$V_{f}=\frac{\frac{b^{\prime }}{M}\cdot d_{a}}{d_{p}\cdot f}$$
Vignetting occurs when $V_{f}$ is larger than 1, as the light cone is cut by the objective lens radius.

### 2.2. Algorithm for 3D Reconstruction

The most important processing steps, necessary to understand the accuracy analysis, are presented here. A detailed explanation on the algorithm has been published by Blaschitz et al. [11]. During the uniform movement of the object, the camera takes *k* pictures. Sequentially, one out of six light sources $C_{s}$ with s=1…6 is used to illuminate the object. Hence, all images taken can be separated into six different image stacks $I_{k,s}$, each of which represents the object illuminated from one particular direction; see Figure 5.

For each image stack $I_{k,s}$, the following processing pipeline is applied:**Feature calculation and multi-view matching:** For each pixel in every image $I_{k,s}$, *m* features are calculated (e.g., *m* = 1 … 3 for 3 different kernel sizes). Multi-view matching is done between the images $I_{k+b,s}$ (forward projection) and $I_{k-b,s}$ (backward projection). Each image $I_{k,s}$ represents the object from a slightly different observation angle. *b* is the baseline distance chosen for the multi-view matching; see Figure 5. This results in *m* confidence and disparity maps for every image $I_{k,s}$. Later, we will investigate the appropriate choice of the baseline *b*. b=1 would mean that matching is done with the neighboring images; hence, a very small baseline but high matching confidence as high similarity between adjacent observations can be expected. If one improves the baseline (increases *b*), the depth estimates will get more accurate, but the individual accuracy of single features will drop due to increased matching ambiguities (also visible in a lower confidence metric) (see also Traxler et al. [1] for definitions of accuracy and precision in this context.)**Fusion of the disparity maps:** To fuse the disparity maps, a mean over the *m* different confidence maps and a weighted mean are calculated (using the individual confidence maps). In the next step, the deviation of the *m* disparities is used to run an additional consistency check by reducing the confidence value in case of discrepancies. This gives exactly one disparity map and one confidence map per image $I_{k,s}$ and baseline *b*.**Generation of the 3D model:** Having employed a rectification method [14] for the inline system, different images $I_{k,s}$ can easily be integrated into a scene illuminated from *s* different light directions; and texture, disparity and confidence for each pixel of the scene, calculated. Due to this calibration, different image stacks stemming from the different illuminations *s* can be integrated, and thus a disparity/confidence tensor can be generated, which is as wide and long as the acquired scene and has “depth” *s* (because of *s* light sources). This means that differently from other methods that construct a depth volume, where “depth” is the number of tested disparity steps, there is a very lean and memory-friendly data structured that actually contains more information, namely, depth and confidence from *s* different illumination and (possibly) view directions, while using less memory.**Stereo baseline:**. In classical stereo imaging, the achievable depth resolution $res_{z}$ is determined by one pixel disparity shift and the stereo baseline used for imaging *b* (see Equation (Equation 3)). In the hypercentric case, the depth resolution is defined by the focal length *f* and the distance between the maximal FOV in scanning direction $b_{max}$ and the projection center distance $d_{p}$ of the chosen aperture *D* (Equation (Equation 5)).**Oversampling:** As our sensor acquires not only the frames of the maximal angle of perspective but also perspectives of frames *n* in-between. Based on oversampling, we assume that additional depth estimation, which is calculated and summed up, increases the accuracy of the depth measurement noise by a factor $\frac{1}{\sqrt{1-n}}$. As described above, the chosen baseline *b* will be projected forward and backward, $I_{k+b,s}$ and $I_{k-b,s}$, and the disparity was calculated. Hence, the achievable depth resolution $res_{zb}$ depends on the baseline *b* used and on the oversampling, dependent on the number of images *n* acquired between $\pm b_{max}$. The final expression for the depth resolution is then Equation (Equation 7).
(7)$$res_{zb}=\frac{p}{2\frac{b}{x_{n}}}\cdot \frac{1}{\sqrt{n-1}}$$$x_{n}$ is the distance of the object to the entrance pupil (see Figure 4).

## 3. Experimental Results

For experimental validation, the methods described in [1] to calculate the depth resolution were used to confirm our theoretical considerations summarized in Equation (Equation 7). A measuring and control surface plate consisting of granite was used, which provides enough structural stiffness and texture to apply our stereo matching algorithm in the desired precision of under 2 μm flatness deviation. For this plate, the 3D shape was reconstructed using the system and methods described above; the temporal measurement noise NM was calculated according to the Fair Data Sheet [12], as described in [1]; from the temporal measurement noise NM using the $\sqrt{8}$, the depth resolution can be calculated following the Fair Data Sheet. To evaluate our postulated baseline and oversampling assumption in Equation (Equation 7), the baseline *b* (distance between frames, where the feature matching is performed) and the oversampling number *n* (frames which are acquired in total) were varied to validate our assumption from above experimentally. Note that for the observation system, the object to be observed must be seen from different perspectives; therefore, a distance of at least three times the maximally achievable baseline $b_{max}$ must be scanned to "see" the full baseline for all points, three times due to our matching process working in forward and backward directions, $I_{k+b,s}$ and $I_{k-b,s}$. Figure 6 shows the results of the experiments with different baselines *b* starting from 0.1 mm up to 1.2 mm. As can be clearly seen in Figure 6, the theoretical considerations (orange line) do not agree with the measured values (blue line). This can be explained by the oversampling factor, which is given by the ratio between the lateral resolution $rex_{xy}$ and the lateral sampling *p*, which is 5.6 for the NA of 0.085. Taking the oversampling factor into account, the results shown in Figure 7 were obtained, which are in agreement with the theoretical results. Only for very low and high baselines do the results deviate significantly from the theoretical result.

With a baseline close to the maximal FOV (1.2 mm), the measured results fluctuate and drop towards zero, as the images to match show entirely different scenes; see Section 2.2. With a baseline of 1.2 mm, the confidence was almost zero for all the measured points. Thus, the larger the baseline, the better the temporal noise, but if too large, no matching points could be found. As to which baseline to use for a good compromise between confidence and temporal noise, experiments were conducted with a set of fixed oversampling rate of 400 frames (see Figure 8). It can be seen that $\frac{1}{3}$ of the maximal baseline seems a good compromise which offers both good temporal noise and still acceptable confidence. With these settings, the depth resolution according to the Fair Data Sheet initiative [12] is 2.8 μm.

## 4. Usecase

The proposed system can be used to scan entire areas at high speeds of up to 12 mm/s using a xy-stage as sample holder.

### 4.1. Laser Cut

A test printing plate containing trenches of various depths, from 350 μm down to 25 μm, provided by OEBS was scanned; see Figure 9. Although having little texture, the system still can reconstruct the surface shape. For this acquisition, the selected stage speed was 4 mm/s and the inter frame step size 62 μm, enabling a scanning time of the whole are of seven seconds. The exposure time was set to 600 μs, and the light level was set to 100% of the maximum, due to low scattering of the plate itself. Hence, such a system can be used as a quality control for laser cuts. The point cloud shown has two scanning lanes.

### 4.2. Klimt Banknote

An artificial banknote provided by OEBS, which shows various security features, including intaglio print, etc., was imaged around the eye of Klimt; see Figure 10. The area was acquired with seven consecutive lines, covering an area of approximately 8 mm × 6 mm. The selected stage speed was 7 mm/s and the inter frame step size 62 μm, enabling a scanning time of the whole area of 20 s. The exposure time was set to 300 μs, and the light level was set to 75% of the maximum. The height of the intaglio print on the eyeball could be precisely measured to be 57 μm. Hence, such a system is perfectly suited for fast security check of very small details. To prove that the system is capable of working with the speed of 12 mm/s, a central line of the Klimt banknote was acquired at 12 mm/s, showing similar quality and details (red dotted box).

## 5. Conclusions

We demonstrated a novel microscopic imaging setup using a hypercentric projection to generate different perspectives that depend on the object’s topology with reference to the focal plane. This effect can be used to calculate stereoscopic 3D topology by acquiring a series of images while an object is translated through the camera FOV. Such systems are ideal for industrial processes, where parts continuously move in a process line. Further, a theoretical treatise for the expected achievable depth resolution and how it depends on the set baseline values and used numerical apertures was given. With the proposed light-field and photometry-based 3D imaging system, scanning speeds of up to 12 mm/s can be achieved with a lateral sampling of 700 nm/pixel, offering a FOV of 1.6 mm. With such a system 3D structures of security features and laser cuts can be visualized and used for fast inline quality inspection and security checks.

## Figures and Tables

**Figure 1 sensors-22-07038-f001:**
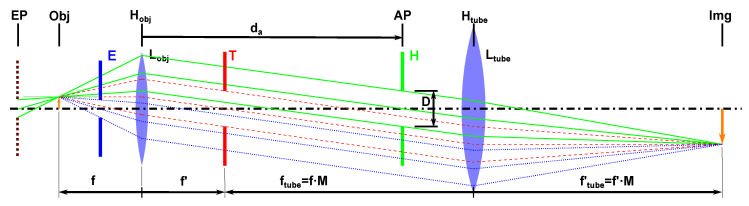
Schematic of a 4f-inspection microscope consisting of an objective lens ($L_{\mathrm{obj}}$), a tube lens ($L_{\mathrm{tube}}$) with their corresponding principal plains ($H_{\mathrm{obj}}$, $H_{\mathrm{tube}}$) and an image sensor (Img). The magnification *M* is defined by the fraction for the tube and objective lens’ focal length ($\frac{f_{\mathrm{tube}}}{f}$). Typically, the aperture is located in the back focal plane of the objective lens (T). In this case, the principal ray in the object space is parallel to the optical axis independent from the object’s height. The entrance pupil is at infinite distance, resulting in the telecentric parallax free projection properties. If the aperture is shifted closer to the object (*E*), the principal ray is deflected towards the entrance pupil location at a finite distance (entocentric projection), resulting in a field-height-dependent observation angle. If the aperture is shifted towards the tube lens (*H*), the entrance pupil (EP), which is defined as the object sided image of the pupil, virtually lays behind the object (hypercentric projection). As the physically acting entrance pupil is only a virtual image, the geometric projection center can be shifted very close to the object without mechanical interference, resulting in a high parallax effect.

**Figure 2 sensors-22-07038-f002:**
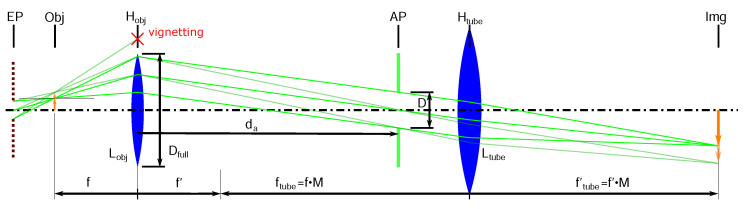
Schematic of a 4f-inspection microscope with hypercentric projection properties. The ray diagram depicts the imaging of two object points with different field heights (two arrows at the position “Obj”). It can be seen that with increasing field height, the principal rays get steeper, producing a stronger parallax effect for the depth estimation in the 3D imaging process. This effect can also be enhanced by increasing the aperture distance $d_{a}$. The maximal usable parallax effect is limited by the opening of the objective lens $D_{full}$, defined by its NA, and the diameter of the observation cone *D*. For the larger object field height, vignetting would occur, which indicates that the $d_{a}$ is too large for the particular configuration.

**Figure 3 sensors-22-07038-f003:**
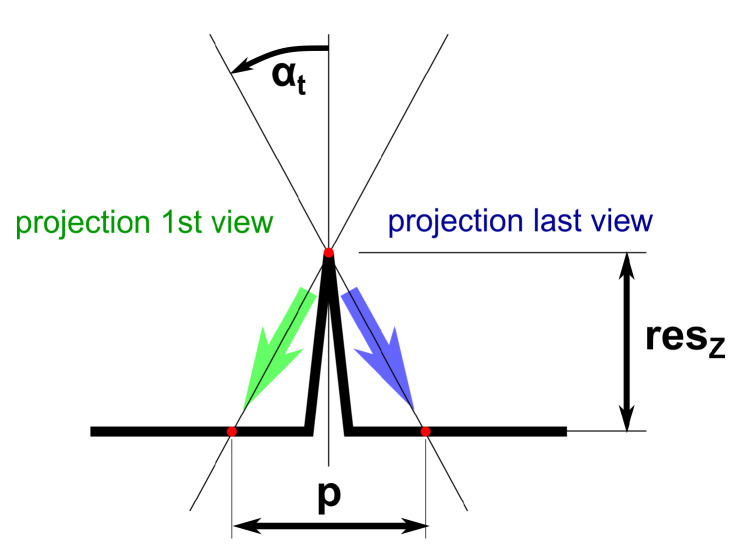
Estimation of the depth resolution. A height step in depth direction $res_{z}$ can be resolved if the parallax between the first and the last view ($\pm \alpha_{t}$) is equal to the lateral sampling *p*.

**Figure 4 sensors-22-07038-f004:**
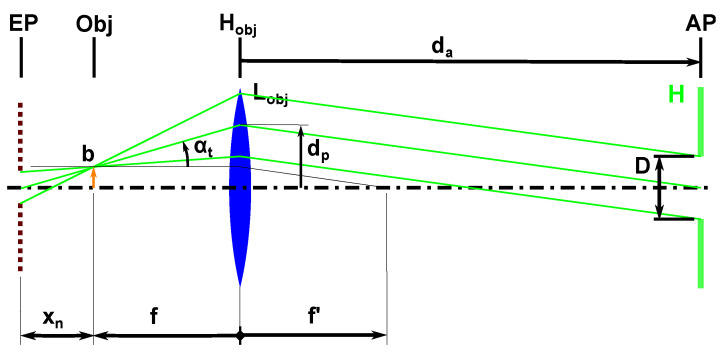
Shows the objects height *b*, the corresponding parallax angle $\alpha_{t}$ and aperture distance $d_{a}$. $x_{n}$ is the distance between the entrance pupil (EP) and the object location.

**Figure 5 sensors-22-07038-f005:**
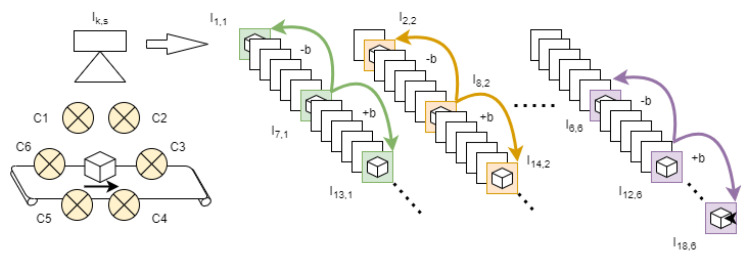
A schematic of the Algorithm, showing the individual images $I_{\left[k,s\right]}$ acquired with the system and the corresponding matching images $I_{\left[k-b,s\right]},I_{\left[k+b,s\right]}$, which are further used to calculate the individual disparity maps, further used for our algorithms.

**Figure 6 sensors-22-07038-f006:**
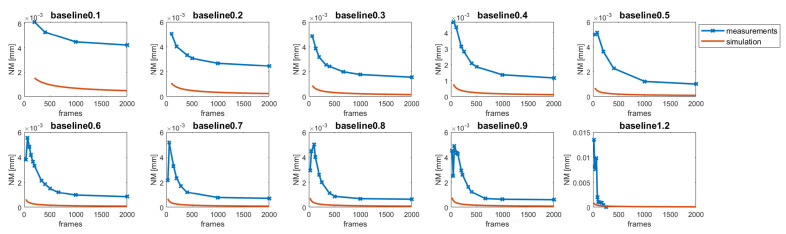
The results of temporal noise (NM) evaluation using the measuring and control surface plate. Different baselines with different oversampling numbers (frames) were chosen. The frames give the number of frames which were used for the 3D reconstruction.

**Figure 7 sensors-22-07038-f007:**
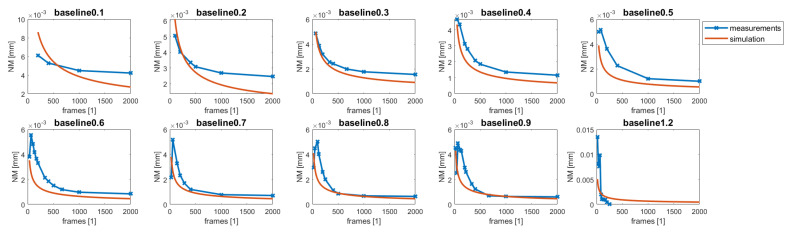
The results of temporal noise evaluation using the measuring and control surface plate using the oversampling factor 5.6 to correct for the fact that the system’s lateral sampling is higher as the actual optical resolution.

**Figure 8 sensors-22-07038-f008:**
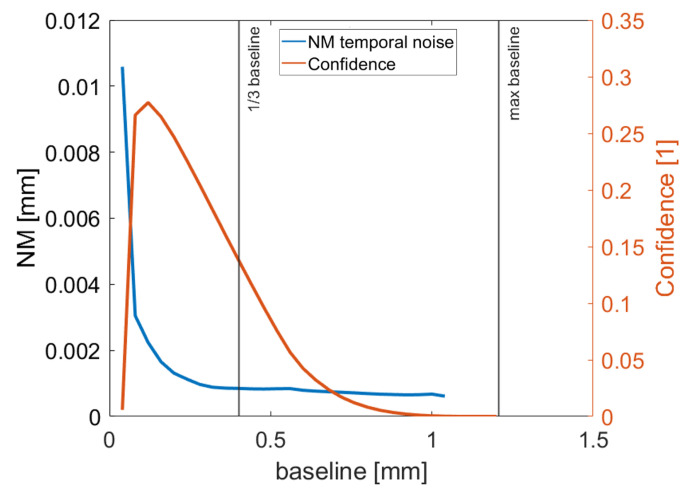
The experimental results of temporal noise NM evaluation and their mean confidence using the measuring and control surface plate using different baselines.

**Figure 9 sensors-22-07038-f009:**
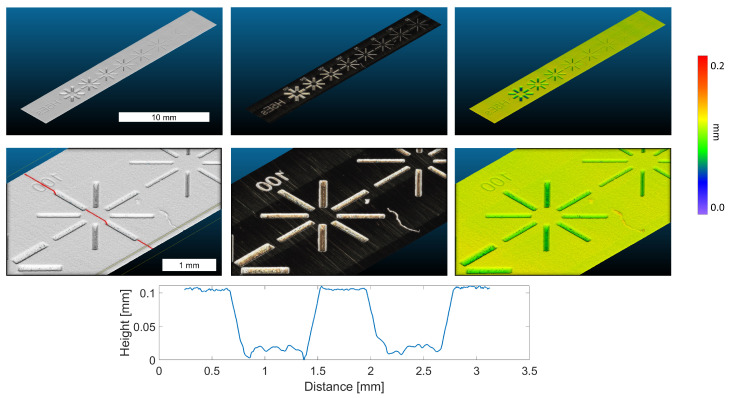
The results of our proposed microscopic imaging setup scanning a print plate generated by using laser cutting. The laser cuts were from 350 μm down to 25 μm in depth. We chose the 100 μm laser cut to measure, as all structures were in our corresponding DOF. As can be seen, despite having only minimal texture, our system could still recover the depth structure. In the lower part of the image, a zoom in on the 100 μm cut is shown with its corresponding depth profile (red line).

**Figure 10 sensors-22-07038-f010:**
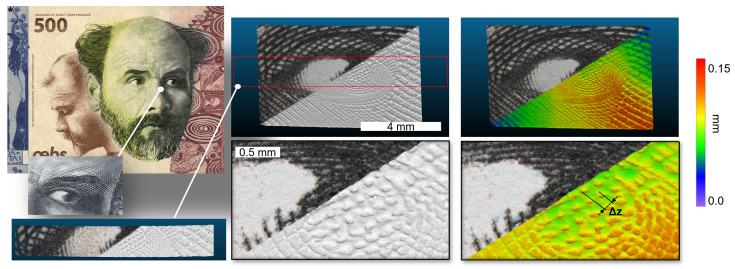
The results of our proposed microscopic imaging setup scanning a Klimt banknote provided by OEBS, with 7 stitched lanes, covering the whole eye. The intaglio print can by nicely seen on the eyeball at the depth measured to be 57 μm. On the left lower side, the result of a single line acquired with 12 mm/s is shown, whereas the 7 lanes were acquired with 7 mm/s.

## Data Availability

Not applicable.

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
