# Peer review of "Fast Inline Microscopic Computational Imaging"

_sensors, 2022, doi:10.3390/s22187038_

Round 1

Reviewer 1 Report

The manuscript “Fast Inline Microscopic Computational Imaging” by Ginner et al. presents an inline microscopic system with hypercentric projection properties for 3D imaging at a depth resolution of 2.8 um and a lateral sampling of 700 nm/pixel, with an acquisition speed up to 12 mm/s. Authors claim that “no other 3D imaging technique exists that offers the same high-resolution lateral sampling as well as acquisition and processing speed”, however, they fail to justify how important the lateral sampling is in the paper. For example, for the imaging people may focus on the lateral resolution of 3.9 um claimed in this paper, which is not superior to that of other techniques mentioned in the introduction. Is fine lateral sampling a key factor to achieve higher depth resolution (2.8 um, better than the lateral resolution of 3.9 um)? Compared authors’ previous work in Ref. 10 and 11, this manuscript fails to emphasize the novelty or the new thing in this paper. It seems to the reviewer the speed is the main improvement, however, there is little information about the high-speed scans except briefly mentioning the “scanning speeds of up to 12 mm/s” in the abstract, introduction, and conclusion.  Below are some additional comments.

There are a number of places hard to read and understand because of inappropriate and incorrect referencing:

1.     Line 90, “the paper will demonstrate a practical industrial use case for inspection of a coin and security features on banknotes”, but in the “Usecase” section laser-cut print plate instead of coin was used.

2.     Line 107, 108, “…such a configuration an object is always seen from above not producing any parallax effect, hence no stereo matching (see above) can be applied.” No sure what is referenced for “seen from above” and “see above”.

3.     In Fig.1 caption, the label “T”, “E”, and “H” are very confusing. For example, “the objective lense” is followed by “(T)”. This should be rephrased, e.g. using “at T position”. plains-> planes

4.     Line 114, in section [4] -> in Section 4?

5.     Line 121-122,  “As objective lens a M = 10x long working distance objective with a NA = 0.28 has been chosen” needs to be rephrased. Should the tube lens have Mx focus length compared with the objective lens?

6.     Between Line136 and 137, what is “see 2.2”? Same issue in Line 153, “see 5”, and Line 215, “scenes 2.2”

7.     In Line 155, what is baseline b here? Is it the same thing as the object height shown in Fig. 4. If it is, why is b=1?

8.     Line 181-183, should it talk about the depth resolution instead of the baseline b?

9.     Line 187, what is n here? Pixel counts, image count,  the refractive index used in Line 134, or the oversampling number n in Line 201.

10.  Line 210, the lateral sampling “pm”?

11.  Figure 6, what does “NM” in the vertical axis mean?

12.  In Figures 9 and 10, figure captions do not match with figures. Please add more details about the experiment and result discussion.

There are still typos and grammatical errors in the manuscript. Please consider further careful revisions. For example:

13.  Line 34, approx.-> approximate, Line 36, 2,5 um-> 2.5 um

14.  Line 70, could demonstrate-> demonstrated

15.  Line 100, “we used a special dome light construction was used”

16.  Line 115, “the pixel size the lateral sampling-”

17.  In Fig.2 caption, “objective lens Dfull )”

18.  Line 139, then-> than

19.  Line 188, “disparity was calculated”

20.  Line 201, “in Eq.7”?

21.  Line 250, e-> We

Last, I would suggest author add a figure of the 3D imaging of the granite control surface plate as this was mentioned in Line 198 “For this plate the 3D shape was reconstructed using the system and methods described above”.

Reviewer 2 Report

This paper aimed to develop a light field based inline microscopic photometry system which achieved 3D imaging with high resolution and high acquisition speed. The method uses a hypercentric projection to generate different perspective views, and computational image fusion techniques. The authors presented a detailed explanation of the imaging system and 3D reconstruction algorithm, which are very helpful to readers. However, descriptions for experimental results (section3) and usecase (section4) may  not be enough to support their conclusions. Therefore, the manuscript should be revised before publication.

1.     It is difficult to understand the section 3. In this section, the authors conclude that the depth resolution is 2.8um, but it is unclear how they derive this result. What is NM in the Figure6? What does the Figure 6,7, and 8 means? The authors need to present these clearly. Besides, how does this imaging system achieve speed 12mm/s? The data supporting this are also required.

2.     The data for section4 is not enough. The authors should add more detailed descriptions of imaging conditions. Moreover, some statistical data which clearly shows usability of the proposed method in a quantitative way are needed.

Author Response

Please see the attachment, comments in green

Reviewer 3 Report

This paper proposed a high-speed 3D imaging method using photometry for continuous manufacturing processes. This paper discussed the 3D imaging with a view to optics, which is different from the conventional phototometry studies based on digital image processing. This study is useful and practical in current inspection processes, and will provide valuable information to the readers. Of course, my decision is accept with minor revision as follows.

1. Questions

- NM[mm] in Figs 6-8 indicates resolution? Does Frames [1] mean the frame number according time or the number of frame to reconstruct a 3D image? Please, clarify the meaning.

- Frames [1] and Confidence [1] in the plot of the results are cited from reference [1]?

- In fig. 8, the legends of the blue and orange curves are required. Your compromised sampling rate was 400Hz at 1/3 base line. However, optimum points are usually determined at maximum of a curve (1/10 base line) or cross points of curves (1/20 or 0.7 base line). Thus, did you have determined the sampling point using a cost function or other method?

2. Others

- Italic font of the text in chapter 4.1 and 4.2

- Descriptions in Fig. 9 and 10 seem to be switched.

- A blank line should be inserted under figure 1 and 2.

- In figure 2, f the objective lens '('D_full)

- for the 113 showcased applications in section [4]. Do you mean section 4?

- I found some missings, such as 'In this paper(,)', 'such a configuration (,)',  'In classical stereo imaging(,)' '(W)e are grateful' and etc. They are usually found in the front of sentences.

- 'a' and 'an' also should be checked, such as 'a(n) object and 'an field height'. 

Author Response

Please see the attachment,comments in green

Round 2

Reviewer 1 Report

I am mostly happy with the authors' reply. However, answers to my questions mostly regarding the importance and novelty of this work are just purely present in the Response to Reviewers. Those should be integrated into the revised manuscript to improve the paper.

There are still typos and grammatical errors in the revised manuscript, please do careful revisions. For example, in Fig. 6 caption, "the number of frames which where  used...". In addition, adding the unit of frames, [1], is very confusing as other reviewers also pointed out. I suggest removing them in both figures and the caption.  

In the "Usecase" section, the figures should be referred in sequence. Authors simply list the scan parameters of stage speed, frame step size, and then give a scan time. First, authors introduce a new parameter of "inter frame step size" which is not introduced in above sections; second, it is unclear how to deduce the scanning time with those parameters. Last, one case used 4 mm/s scan speed and the other 7 mm/s, both of which are below 12 mm/s. Therefore, it is not convincing to claim that the scan speed is up to 12 mm/s. I would suggest change the statement if there is no data to support this claim.

Author Response

Please see the attachment, comments in green, thank you for the detailed review!

Reviewer 2 Report

The manuscript has much been improved in the second version. It is thus ready for publication. 

Author Response

Thank you very much for your time and effort to review our paper!
best Laurin